Understanding mental fatigue and its detection: a comparative analysis of assessments and tools

Kunasegaran Kaveena 1
Ismail Ahamed Miflah Hussain 2
Ramasamy Shamala 1
Gnanou Justin Vijay 3
Caszo Brinnell Annette 4
Chen Po Ling 2 chen.poling@nottingham.edu.my
1 Department of Psychology, International Medical University , Bukit Jalil, Kuala Lumpur , Malaysia
2 School of Psychology, University of Nottingham Malaysia , Semenyih, Selangor , Malaysia
3 Department of Biochemistry, International Medical University , Bukit Jalil, Kuala Lumpur , Malaysia
4 Department of Physiology, International Medial University , Bukit Jalil, Kuala Lumpur , Malaysia
Aleman André
Electronic publication date: 2023 Aug 23
Publication date: 2023
Volume: 11
Electronic Location ID: e15744
Received 2023 Feb 9; Accepted 2023 Jun 21
Copyright: © 2023 Kunasegaran et al.
Copyright year: 2023
Copyright holder: Kunasegaran et al.
License: This is an open access article distributed under the terms of the Creative Commons Attribution License, which permits unrestricted use, distribution, reproduction and adaptation in any medium and for any purpose provided that it is properly attributed. For attribution, the original author(s), title, publication source (PeerJ) and either DOI or URL of the article must be cited.
License URL: https://creativecommons.org/licenses/by/4.0/

Keywords: Mental load, Saccades, Executive function, Attention, Digital health, Mental health

Funding: Fundamental Research Grant Scheme (FRGS) FRGS/1/2020/SS0/IMU/02/1 This work was supported by the Fundamental Research Grant Scheme (FRGS), Grant Number FRGS/1/2020/SS0/IMU/02/1. The funders had no role in study design, data collection and analysis, decision to publish, or preparation of the manuscript.

==============================
Mental fatigue has shown to be one of the root causes of decreased productivity and overall cognitive performance, by decreasing an individual’s ability to inhibit responses, process information and concentrate. The effects of mental fatigue have led to occupational errors and motorway accidents. Early detection of mental fatigue can prevent the escalation of symptoms that may lead to chronic fatigue syndrome and other disorders. To date, in clinical settings, the assessment of mental fatigue and stress is done through self-reported questionnaires. The validity of these questionnaires is questionable, as they are highly subjective measurement tools and are not immune to response biases. This review examines the wider presence of mental fatigue in the general population and critically compares its various detection techniques (i.e., self-reporting questionnaires, heart rate variability, salivary cortisol levels, electroencephalogram, and saccadic eye movements). The ability of these detection tools to assess inhibition responses (which are sensitive enough to be manifested in a fatigue state) is specifically evaluated for a reliable marker in identifying mentally fatigued individuals. In laboratory settings, antisaccade tasks have been long used to assess inhibitory control and this technique can potentially serve as the most promising assessment tool to objectively detect mental fatigue. However, more studies need to be conducted in the future to validate and correlate this assessment with other existing measures of mental fatigue detection. This review is intended for, but not limited to, mental health professionals, digital health scientists, vision researchers, and behavioral scientists.

Introduction

In today’s society, individuals at different stages of their lives (e.g., students and working adults) constantly experience mental load within their day-to-day activities. Taking working adults as an example, their experience of mental load begins when individuals wake up in the morning and check their phones for new messages or emails. As the day goes by, they are faced with complex tasks at work such as organizing, planning and managing, in addition to unexpected interruptions such as attending to a colleague. Mental load is further aggravated upon returning home from work, to deal with household and personal responsibilities such as cleaning, taking care of children and grocery shopping. The continuous experience of mental load results in an individual feeling mentally fatigued, stressed, and anxious about the extensive list of unfinished tasks for the day. Occasionally, we read headlines about doctors or nurses involved in traffic accidents due to lack of sleep or being overworked. Statistics from Australia show that 10–40% of traffic accidents result from driving fatigue caused by mental fatigue (Victorian Transport Resources, 2019). However, the cause of fatigue is often not driving, but the mental load that they have experienced before they begin to drive, such as vigorous mental tasks of crunching numbers as an accountant or the heavy workload of performing a 12-h surgery as a heart surgeon (Victorian Transport Resources, 2019). Mental fatigue has an impact on various age groups and can lead to fatal accidents putting fatigued individuals and bystanders in danger.

Unfortunately, working adults are not the only ones susceptible to mental fatigue. Many students in universities sit through 4–6 h of lectures a day and spend 2–3 h outside of the classroom revising. High school students in America spend an average of 6 h a day in classes (Jin, 2012). Often, a student will proceed from one class directly to another with no breaks in between. These long schedules exist in educational settings, despite studies showing that students experience a drop in attention after 10–15 min from the start of a lecture (Bradbury, 2016). Learning under stressful conditions (i.e., long hours with saturated amounts of information) has shown to increase an individual’s mental load (Gaillard, 1993). Sitting in classes for long hours despite a student’s short attention span increases the student’s mental load resulting in many of them feeling mentally fatigued by the end of their classes (Petrov et al., 2020).

Mental fatigue is further complicated by stress. Stress has a potential fatigue-enhancing effect when an individual is performing a highly cognitive- and attention-demanding task. Furthermore, stress can have an impact on learning because it directly affects brain areas that control for working memory, increasing the risk of cognitive overload (Bong, Fraser & Oriot, 2016). Therefore, it enhances mental fatigue causing individuals to spiral into a state of depression and other mental health and social issues (i.e., suicide and anxiety) (Craik, 2014; Tanaka, Ishii & Watanabe, 2014; Forster & Lavie, 2016). Recent numbers of suicide cases have increased due to the global pandemic that started in 2019 (Sher, Peters & Sher, 2020), and with the suicide rates of physicians 44% higher than the general population due to burn out and mental exhaustion (Reger, Piccirillo & Buchman-Schmitt, 2020), it would be beneficial for workplaces to be able to detect mental fatigue in their employees. As many companies and educational institutes resort to a work or study from home strategy in hopes of keeping their business and education progress afloat during the pandemic, the nature of finding a good work-life balance has taken a drastic shift putting many individuals under increasing amounts of stress (Charles et al., 2021). Mental fatigue can be detrimental and affects individuals of varying backgrounds and age, be it a working adult or a young teen in the university. The impacts of mental fatigue warrant a comprehensive evaluation and comparison of various (subjective and objective) mental fatigue detection and assessment tools, and their strengths and weaknesses, but no single study in the literature has addressed this. This serves as the main rationale that motivates the current comparative analysis. In this analysis, the impact of mental fatigue in the general population is also discussed along with some of the current detection methods and its implementation in health monitoring technologies.

Mental fatigue

Mental fatigue and physical fatigue are two separate entities. Physical fatigue is a form of tiredness caused by repeated muscle movements (Mizuno et al., 2011). In contrast, mental fatigue is defined as a psychobiological state of tiredness caused by prolonged periods of performing demanding, cognitive-load-inducing activities, and it reduces efficiency in cognitive performance (Craik, 2014; Tanaka, Ishii & Watanabe, 2014). The concept of mental load can be explained by perceptual load and processing capacity. First, perceptual load is associated with the amount of information involved in processing a relevant task (Forster & Lavie, 2016). High perceptual load results in inattentional blindness leading to impairments of memory (Murphy, Groeger & Greene, 2016; Murphy & Greene, 2016). Perceptual load is one of the causes of high mental load; however, higher mental load can also occur from tasks involving executive functioning skills irrespective of whether or not they involve higher perceptual load (Blair, 2016). Second, processing capacity can be defined as the amount of information an individual can process at any given time (Hancock et al., 2021; van Elburg et al., 2021). Once the capacity is reached, the ability to process new information becomes difficult, resulting in mental overload. Experiencing long durations of mental overload leads to mental fatigue (Mizuno et al., 2011).

Differences between stress and mental fatique

Stress and mental fatigue can be differentiated based on their inducers (Gaillard, 2001). The mechanisms underlying stress and mental fatigue are different and produce bio-behavioral states that are distinguishable. Stress is induced when a stressor (a perceived threat) causes the body to release stress hormones that result in behavioral and physical changes (Lupien et al., 2009; Reznikov et al., 2007). Unlike mental fatigue, stress does not result in a feeling of tiredness. Instead, it causes the body to produce a fight-or-flight response when experiencing stress (Johnson et al., 1992). It arouses the central nervous system (CNS) or the sympathetic nervous system (SNS) to elicit different stress responses depending on the situation (Adams et al., 1969; Schneiderman, Ironson & Siegel, 2005). The biological response of stress causes the body to release hormones such as cortisol and adrenaline which helps activate the body to deal with a perceived threat. However, when the body is activated with stress hormones for a longer period of time, it begins to disrupt normal body functions such as cognitive performance, sleep and digestion (Chu et al., 2021; Yaribeygi et al., 2017). Studies have shown that long term stress can cause mental fatigue as well as structurally modify different parts of the brain (Mariotti, 2015; Kocalevent et al., 2011). Unlike stress, mental fatigue does not accompany the feeling of panic, frustration, or the pressure to succeed and produce a flight-or-fight response (Adamsson & Bernhardsson, 2018). Mental fatigue causes a feeling of mental tiredness that results in decreased attention and impairments in executive functions. It arises when an individual experiences continuous mental load unlike stress that could only require a single stimulus to be produced (Kocalevent et al., 2011).

Active and passive fatique states

Desmond & Hancock (2001) were the first to propose two forms of mental fatigue states: “active state” of fatigue (i.e., mental fatigue induced by prolonged task-related perceptual-motor adjustments) and “passive fatigue” (i.e., when a person becomes mentally fatigued from prolonged periods of inactivity such as driving a car). Due to its monotonous nature, passive fatigue causes sleepiness and a decrease in attention without the typical symptoms of sleepiness (e.g., yawning) (He et al., 2023; Hu & Lodewijks, 2020). Different discriminative functional connectivity features of the brain have been observed when individuals are in passive and active mental fatigue states, suggesting that functional connectivity could be used as a biomarker for mental fatigue distinction and detections (Dimitrakopoulos et al., 2018). Bernhardt et al. (2019) investigated the differences between the two forms of fatigue using electroencephalographic (EEG) indices. They found that individuals experiencing passive fatigue showed significantly less EEG engagement in the parietal-occipital midline (as indicated by a decreased alpha power spectral density caused by a loss in task engagement) compared to those experiencing active fatigue. Distinct saccadic eye movement characteristics have also been observed in individuals experiencing active mental fatigue (e.g., slower blinks) and passive mental fatigue (e.g., significantly decreased pupil dilation) (Desmond & Hancock, 2001). However, brain mechanisms that are responsible for inducing the two different states have not been investigated. In this review, a focus will be placed on the active state of fatigue due to its likelihood to be induced by constant workload that depicts a regular busy day in this modern era.

Environmental causes of mental fatique

Mental fatigue is often caused by environmental factors such as high-pressure occupations, long working hours and job strain. Mental fatigue arises when an individual is exposed to lengthy periods of tasks or activities that require the use of executive functions such as working memory, flexible thinking, and self-control (Diamond, 2013). High-pressure occupations such as surgeons and air traffic controllers experience mental fatigue from increased mental workload for prolonged amounts of time (Diamond, 2013; Nealley & Gawron, 2015). Surgeons often have to conduct surgeries that could take up to 12 h, depending on the complexity of the surgery. Attention and concentration are constantly required to perform well in such occupations, but they cause mental fatigue. In addition, long working hours can cause general fatigue (physical and mental). Studies have shown that working more than 280 h a month causes an increase in general fatigue, physical disorders, anxiety, and chronic tiredness (Petrut et al., 2020; Nagashima et al., 2007). These studies also showed that a maximum of 260 h a month is ideal to minimize fatigue symptoms. Despite this, individuals lacking financial resources tend to work more than the recommended hours a month to avoid poverty (Bick, Fuchs-Schündeln & Lagakos, 2018). In a study conducted on sugar cane harvesting workers in Brazil, 17.9% of the workers mentioned mental fatigue as a challenge living in poverty (Rocha, Marziale & Robazzi, 2007). The mental fatigue mentioned stemmed from frenetic work rhythms, constant attention and concentration, monotony, repetitiveness, and threat of unemployment. Moreover, a descriptive survey on orphaned and vulnerable children stated that overworking and fatigue was a major challenge for the children surviving in deep poverty (Ganga & Chinyoka, 2010).

Job strain, which is another cause of mental fatigue, occurs when there is an imbalance between strict job requirements and an inadequate ability to adapt and cope. Job strain is known as a form of stress caused by high demands and the inability to control or contribute to decisions that commonly occur at workplaces (Shakerian, Habibi & Poorabdian, 2015). By using structured questionnaires, it has been found that individuals experiencing elevated levels of job strain are mentally fatigued (Perrier et al., 2016; Tran et al., 2020). This leads individuals to then experience what is called occupational burnout (Ekstedt et al., 2006). Occupational burnout can lead to working adults developing mental health problems such as depression, causing many individuals to spiral into suicidal thoughts and actions (Ahola et al., 2014).

Neural mechanisms of mental fatique

The frontal and occipital cortical regions of the brain have been primarily associated with mental fatigue. Based on studies measuring EEG responses, slow wave activity increases over the entire cortex in mentally fatigued individuals (Craig et al., 2012; Qi et al., 2019). These changes were most evident in theta and alpha brain waves, and are dominant in frontal, central and posterior lobes of the brain (Craig et al., 2012; Perrier et al., 2016; Tran et al., 2020). The changes observed in the band waves’ activities suggest that as mental load increases, the brain loses its capacity to cope with the mental loading, thus slowing down brain activity (Klimesch, Sauseng & Hanslmayr, 2007). This results in individuals having decreased alertness and decreased executive functions. Moreover, it has been found that individuals that were mentally fatigued from performing cognitive tasks, produced decreased alpha-frequency band (8–13 Hz) power in the middle occipital gyrus (i.e., in the region responsible for object recognition), cuneus, and middle temporal gyrus (Brodmann’s area 19 and 39) which is responsible for language and semantic memory processing, visual perception, and multimodal sensory integration (Tanaka, Ishii & Watanabe, 2015). Although most evidence relates reduced brain activity to mental fatigue, Tanaka et al. observed increased beta frequency bands in the frontal cortex using magnetoencephalography, while individuals performed a mentally-fatiguing task (Tanaka, Ishii & Watanabe, 2014). Additionally, a recent study using functional near-infrared spectroscopy has also reported strong functional connectivity within prefrontal regions, particularly for severe mental fatigue conditions (Peng et al., 2022). To reconcile, changes in brain activity observed in the frontal and occipital cortices suggest that mental fatigue affects these areas of the brain. However, whether mental fatigue is characterized by enhanced or suppressed brain activity (or both) needs to be further examined to gain better insight.

The dual regulation system of mental fatigue is a conceptual model that describes the neural mechanisms of mental fatigue. It suggests that mental fatigue is a product of mental facilitation and inhibition systems (Ishii, Tanaka & Watanabe, 2014). According to this system, the mental facilitation system is activated to maintain performance on a task that induces mental workload whilst activation of the mental inhibition system impairs performance on tasks that induce mental workload when the processing capacity of an individual has reached its limit. The brain areas involved with the mental facilitation system are the thalamic-frontal loop that interconnects the limbic system, basal ganglia, thalamus, and orbitofrontal cortex, whilst the mental inhibition system constitutes the insular cortex and posterior cingulate cortex (Mohammadzadeh, Farsi & Khosrowabadi, 2020). Taken together, the frontal cortex and visual cortices are brain areas that are strongly affected by mental fatigue with the mental facilitation and inhibition systems modulating the cognitive performances of an individual (Sabeti, Boostani & Rastgar, 2018; Faber, Maurits & Lorist, 2012).

Effects of mental fatigue

Cognitive performance

Research has shown that mentally fatigued individuals tend to have decreased performance in cognitive tasks that require vigilance, sustained attention, and divided attention (Smith et al., 2019). On the one hand, this observation could be partially attributed to fatigue-related motivation. Motivation has been found to play a role in the declining cognitive performance observed in individuals that are mentally fatigued. A study comparing cognitive performance found that individuals that were not being rewarded showed less cognitive effort, decreased frequency of saccades and increased blinks compared to participants that were expecting a reward when both groups reported being mentally fatigued (Herlambang, Taatgen & Cnossen, 2019). Similarly, it was found that individuals with higher levels of intrinsic motivation performed better in a cognitive task compared to individuals with low levels of intrinsic motivation regardless of being mentally fatigued or not (Herlambang, Cnossen & Taatgen, 2021). Nevertheless, there are limited objective measures to firmly associate motivation declination caused by mental fatigue to be the core factor of a poorer cognitive performance (Jimura, Locke & Braver, 2010).

On the other hand, Hopstaken et al. (2016) and Gergelyfi et al. (2015) stated that a depletion in performance is not due to reduced motivation in an individual but due to the lack of mental resources available to efficiently process and engage in the task. The prefrontal cortex has a critical role in producing goal-directed signals for attention switching (Rossi et al., 2009). It has been shown that when an individual is mentally fatigued, an overactivation occurs in the frontal cortex that results in an inhibition to process information (Tanaka, Ishii & Watanabe, 2014; Ishii, Tanaka & Watanabe, 2014; Goodman & Marino, 2020). This explains the cause of decreased executive control (e.g., inhibition responses) and an increase in response time when performing a task (Gantois et al., 2020; Guo et al., 2018; Jaydari Fard, Tahmasebi Boroujeni & Lavender, 2019). As a result, individuals with mental fatigue show decreased cognitive performance (Sasahara et al., 2015) and are less likely to be able to detect hazards which could risk occupational safety (Rocha, Marziale & Robazzi, 2007). Individuals experiencing mental fatigue find it difficult to suppress irrelevant information (Faber, Maurits & Lorist, 2012; Csathó et al., 2012). Goal-directed action is inhibited when a person is mentally fatigued, resulting in stimulus-driven actions to be more pronounced (Boksem, Meijman & Lorist, 2005). To be brief, it is harder to inhibit stimulus-driven actions when an individual is mentally fatigued, resulting in an increase in reaction times and error rates when performing tasks that require an individual to actively suppress irrelevant information. This phenomenon reflects the existence of decreased selective visual attention due to mental fatigue (Guo et al., 2016).

Physical performance

Mental fatigue has also shown to impair physical performance such as balance control when attentional processing is compromised (Webster, Richter & Kruglanski, 1996). Pires et al. (2018) showed that mentally fatigued individuals have impaired physical performance (e.g., balance control, physical strength) that was also associated with an increased activation in the prefrontal cortex, as reflected in increased slow theta brain waves, resulting in increased bottom-up responses and decreased top-down processes. Similarly, Marcora, Staiano & Manning (2009) stated that mentally fatigued individuals were found to have higher perceptions of effort when performing physical activities which resulted in individuals reaching their perceived exertion limits faster, and therefore ending their physical activities. In support of this finding, Penna et al. (2018a) also showed that a decline in physical performance in mentally fatigued individuals was not attributed to physiological variables such as oxygen uptake, heart rate or blood lactate, but can be attributed to higher perceived exertion which is an effect of mental fatigue. Mental fatigue does not affect the maximum strength and power that can be exerted by an individual but rather it affects the perception (i.e., effort exerted, or effort required to complete the task) of an individual which leads to a decrease in physical performance (Van Cutsem et al., 2017).

Mental fatique detection

Health monitoring technologies are used to monitor activities of daily living, cognitive decline, mental health, heart conditions, and many more (Liu et al., 2016). The use of similar health monitoring technologies in workplaces can allow companies to easily detect mentally fatigued individuals. With mental fatigue drastically affecting everyday life, there is an increasing demand for more robust detection methods to identify it early, before they cause too much damage. Given that mental fatigue is a non-specific symptom, making it difficult to identify the origins of the condition, a singular parameter of evaluation would not be sufficient. A combination of detection methods must be used to validate a single parameter. As discussed below, a variety of methods have been used in the past to detect mental fatigue such as self-reporting questionnaires, the Attention Network Test (ANT), EEG, heart rate variability (HRV), salivary cortisol levels and saccadic eye movements.

Self-reporting questionnaires

Various self-assessments have been developed in the pursuit of mental fatigue detection. The diverse types of questionnaires that have been developed to measure mental fatigue are summarized in Table 1 along with their individual strengths and weaknesses. In general, subjective measures such as self-reporting questionnaires have often been criticized to be biased as the honesty (i.e., being truthful instead of answering based on socially acceptable behaviors) and introspective abilities (i.e., to accurately assess themselves) of participants can be easily compromised. Humans are also unable to accurately make judgments regarding their cognitive states (Schmidt et al., 2009). This is why experts strongly suggest that these questionnaires be used hand in hand with other objective measures (Tempelaar, Rienties & Nguyen, 2020). This brings to surface a limitation as it does not provide a quick and simple method of detecting transient states of mental fatigue. These questionnaires are unable to provide moment-to-moment fluctuations of mental fatigue and often these methods are affected by cognitive ability, mood, and anxiety levels.

Table 1 Summary of self-reporting questionnaires used to detect mental fatigue.

Name of questionnaire	Description	Strengths	Weakness	
Occupation fatigue exhaustion recovery (OFER) scale (Winwood et al., 2005)	15 items, classify and distinguish three subscales of mental fatigue (i.e., chronic work-related fatigue traits, acute and end-of-shift states, and effective fatigue recovery between shifts).	Robust and embodies a gender bias and work-status free psychometric characteristic.	Only detects three domains of mental fatigue.	
Visual analog scale to evaluate fatigue severity (VAS-F) (Lee, Hicks & Nino-Murcia, 1991)	18 items, l00-mm lines in the case of visual analogue lines between two extremes “not at all tired” to “extremely tired”.	Uses semantic differential scale which gives a unique bipolar ordinal scale format that captures a person’s feelings about a given item.	Frequent reluctance of individuals to use the highest and lowest extremes.	
Fatigue assessment scale (FAS)	10-items, evaluates symptoms of chronic fatigue. Half the items measure physical fatigue and the other half measures mental fatigue. It is a unidimensional scale measuring fatigue independently from depression (Michielsen, De Vries & Van Heck, 2003).	It was found to be the most promising fatigue measure when compared to five other fatigue questionnaires (De Vries, Michielsen & Van Heck, 2003).	Four items appeared to be gender bias—women tended to score significantly higher than men (De Vries et al., 2004).	
Multidimensional fatigue inventory (MFI) (Smets et al., 1995)	20 items, dimensions covered: General Fatigue, Physical Fatigue, Mental Fatigue, Reduced Motivation and Reduced Activity.	Good internal consistency (Cronbach’s Alpha = 0.84).	Has not been validated with mentally fatigued individuals.	
Fatigue severity inventory (FSI)	14 items, evaluates multiple aspects of fatigue such as perceived severity, frequency, and interference with daily functioning (Donovan & Jacobsen, 2011).	Validated with both female and male cancer patients with an age range of 18–24. It has an internal consistency of 0.94 (Hann, Denniston & Baker, 2000).	Does not correlate with mental fatigue in the MFI (Lou et al., 2001).	

Attention Network Test (ANT)

Another method proposed to detect mental fatigue is the use of the ANT (Fan et al., 2002). This test basically examines the attentional network of an individual in three fundamental areas such as alerting, orienting, and executive control (Federico et al., 2013). The alerting network is examined by the changes in reaction times insinuated by a warning signal. The orienting network is evaluated by the changes in reaction times resulting from cues indicating where the target will occur. The efficiency of the executive network is examined by requiring the subject to respond by pressing a key indicating the direction of a central arrow surrounded by congruent, incongruent, or neutral flankers. This method proposes that individuals that are mentally fatigued will perform worse than a non-fatigued individual, as their attention is compromised by mental fatigue (Lou, 2009). Research has shown that mentally fatigued individuals display significantly lower executive and attention network efficiency and accuracy compared to controls (Pauletti et al., 2017; Togo et al., 2015).

The implementation of the ANT in health monitoring technologies are not promising as Jones et al. (2016) found that a practice effect was common in healthy adults when the test is repeated after a 24-h interval, suggesting that test-scores can be affected if this detection tool were to be used as a daily method of assessing mental fatigue. Moreover, attention systems comprise various independent components making it too complicated to be evaluated by a single task (Wang et al., 2015). For example, attention consists of four independent systems (i.e., endogenous, exogenous, spatial, objective) that vary in behavioral effects and have specific neural substrates (Chica, Bartolomeo & Lupiáñez, 2013; Weinbach & Henik, 2012). Furthermore, this detection tool is yet to be validated against objective and/or subjective measures (e.g., salivary cortisol levels, self-reporting questionnaires, saccadic eye movements, etc.) of mental fatigue. The testing duration of ANT is also relatively long as it takes an average of 25 to 30 min, depending on the length of the test. The lengthy testing period raises potential compliance and burden issues (Kas, Serretti & Marston, 2019). Also, although this method of detection is not as subjective as questionnaires’ responses and able to provide objective measurements (e.g., quantifiable data from pressing a key), the ANT task is more of a cognitive assessment which is mainly used for inferring purposes (Kayser et al., 2020). This may lack clarity and direct observation that potentially lead to questions about its validity in detecting mental fatigue.

EEG

Mental fatigue can also be measured using EEG spectral changes in the theta, alpha and beta waves as summarized in Table 2 (Cheng & Hsu, 2011). In a recent study, theta wave activity was found to be a promising primary biomarker for mental fatigue and the alpha wave activity was considered a second line biomarker (Tran et al., 2020). Studies have shown that the use of EEG recordings alongside electrooculography (EOG; used to assess the cornea-retinal standing potential that exists between the front and the back of the eye) increases the accuracy in detecting mental fatigue (Jia & Tyler, 2019). This method has been reported to provide most accurate measures of mental fatigue. In contrast, Smith et al. (2019) found that cognitively demanding tasks used to induce mental fatigue in individuals had unclear effects on EEG results. Moreover, the major limitation of using EEG as a mental fatigue detection tool is that it is too sophisticated and requires a laboratory setting to minimize signal to noise ratio (Read & Innis, 2017). A robust assessment method would be needed in order to ensure this method is able to handle the variations in electric signal between individuals that occur due to the EEG signal being highly sensitive (Monteiro et al., 2019). EEG also requires the use of electrodes which needs to be placed specifically on an individual’s head in order to get accurate measurements of the alpha, beta, and theta wave rhythms, making it perplexed to integrate into a simple health monitoring device such as a smartphone and surveillance devices that would be able to monitor individuals’ mental fatigue continuously whilst they carry out their daily tasks. This problem has been addressed in several studies, whereby researchers developed a simple measurement technique that is just as effective as the entire EEG setup with the use of only one EEG sensor to detect mental fatigue, thus decreasing the complexity of this detection method (Resalat & Saba, 2015; Rohit et al., 2017; Hu, 2017). A recent study on the impact of mental fatigue on brain activity showed that when using the entire EEG setup, the EEG relative power index was better at assessing mental fatigue when individuals were in a resting state compared to when they were actively performing a task (Li et al., 2020). Overall, this method has the ability to reflect the changes in the brain associated with mental fatigue, however in light of recent findings (Smith et al., 2019), it has been shown that this method may not be able to accurately detect the changes that occur in the brain whilst a mentally fatigued individual is carrying out a cognitively demanding task which would limit the use of this detection tool. As individuals tend to go for long hours before taking a break, and to avoid prolonged mental fatigue go unnoticed, it would be more beneficial to have mental fatigue detected during task engagement as an indicator to needing a break before continuing to the next task.

Table 2 Type of brain waves measured by EEG.

Type of brain waves	Frequency (Hz)	Description	Changes that occur when mentally fatigued	
Beta	14–30	Awake, normal alert, conscious
High/fast beta waves occur when in complex thought process. Slow/low beta waves occur when pondering over something.	Increased frequency bands in the frontal lobe (Tanaka, Ishii & Watanabe, 2014).	
Alpha	9–13	Active when physical and mentally relaxed, awake but drowsy.	Increased slow wave activity in the central, frontal and posterior lobes of the brain (Craig et al., 2012). Decreased frequency bands in middle occipital gyrus, cuneus, and middle temporal gyrus (Tanaka, Ishii & Watanabe, 2015).	
Theta	4–8	Active when deeply relaxed or in an autopilot state. This wave is important for processing information and making memories.	Increased slow wave activity in the central, frontal and posterior lobes of the brain (Craig et al., 2012).	

Heart rate variability (HRV)

Apparies, Riniolo & Porges (1998) reported this cardiovascular index to be a sensitive measure of fatigue and mental workload. Heart rate is controlled by the autonomic nervous system (e.g., SNS) whereby the release of hormones (e.g., Estrogen) causes fluctuations in heart rate (Leicht, Hirning & Allen, 2003). These fluctuations occur according to the ratio between the absolute low frequency and high frequency power measured via HRV (Goldberger, 1999). HRV is a simple, non-invasive measure of the changes between each heartbeat in respect to time, using an electrocardiogram (ECG) (ChuDuc, NguyenPhan & NguyenViet, 2013). HRV also reflects the increase of the body’s self-regulatory abilities (e.g., behavioral, cognitive and emotional process) when performing mentally fatiguing tasks. The body’s self-regulatory processes (specifically the cognitive processes) play an important role in changing or suppressing information (Barutchu et al., 2013; Heatherton, 2011). Segerstrom & Nes (2007) found an increase in HRV during high self-regulatory efforts. Various studies have reported similar findings, that an increase in heart rate is strongly correlated with mental fatigue (Huang et al., 2018; Delliaux et al., 2019; Melo, Nascimento & Takase, 2017). These findings were supported by EEG data—a decrease in P300 amplitude indicative of a decrease in attention correlated with an increase in heart rate (Zhang & Yu, 2010; Zhao et al., 2012).

The HRV detection tool has been integrated into smart devices (i.e., smart watch, fitness bracelets) to be used as health monitoring technologies (Naranjo-Hernández et al., 2017). It is mainly used to measure heart rate in real time for the purpose of cardiovascular health monitoring but has not been successfully adapted to specifically detect mental fatigue. Nelesen et al. (2008) investigated the relationship between fatigue and heart rate, finding that fatigue was not associated with heart rate but rather significantly associated with decreased cardiac index and stroke index when at rest. Accordingly, the reliability of detecting mental fatigue via HRV is heterogeneous with various studies demonstrating no link between mental fatigue and physiological variables such as heart rate (Penna et al., 2018b; Thompson et al., 2019).

In addition, HRV can be influenced by too many confounding variables such as movements made by individuals during the assessment resulting in false readings, pathological conditions that affect the nervous system (e.g., degenerative diseases, brain damage), the heart (e.g., arrhythmia), or other organs (e.g., renal failure), mental health disorders (e.g., depression), smoking, caffeine, and medication (Acharya et al., 2006; Hayano et al., 1990; Monda et al., 2013; Massaro & Pecchia, 2016). Although it would make sense for physical fatigue to be linked with heart rate, there are no direct links of heart rate to mental fatigue. Gonzalez et al. (2017) stated that the current algorithm used for HRV detection via smartwatches and simple heart rate detection devices are unable to distinguish the differences between mental fatigue and physical fatigue. Thus, HRV may not be sufficient for the use of mental fatigue detection.

Cortisol levels

Mental fatigue has been found to modulate the autonomic and endocrine responses in the body (Moreira et al., 2018). Fatigued individuals precipitate an increase in cortisol levels in the body released by the hypothalamic-pituitary-adrenal axis (Torres-Harding et al., 2008; Thompson et al., 2016). The increase of cortisol levels has been associated with a decrease in cognitive functions, specifically attention, after a fatigue-inducing task (Bohnen et al., 1990; Lee et al., 2007). Salivary cortisol levels have been shown to be a reliable indicator of mental fatigue because they reflect blood cortisol levels, which rise when an individual is stressed (Kirschbaum & Hellhammer, 2007).

In contrast, Powell et al. (2013) found that total cortisol output was scarcely associated with fatigue. Another limitation of this method is that not all of the studies mentioned above had examined the relationship between cortisol level and mental fatigue specifically. Most of these studies evaluated physical and mental fatigue as a whole, rather than studying mental fatigue separately. For example, in Torres-Harding et al. (2008), where a relationship between cortisol levels and the symptoms of chronic fatigue syndrome was observed, the fatigue associated with the salivary cortisol levels was not specifically related to mental fatigue but more so to fatigue in general. As mentioned above, physical and mental fatigue are two different forms of fatigue that need to be evaluated separately as one can experience fatigue physically but not mentally and vice versa. Hence, there is limited data that can specifically link changes in cortisol levels to mental fatigue. Furthermore, a recent review reported a lack of evidence that salivary cortisol is correlated with stress and fatigue, suggesting that the relationship between mental fatigue and cortisol levels remains inconclusive (de Assis, de Resende & Marziale, 2018).

Nonetheless, a combination of salivary cortisol measures and health monitoring technology seems to be remarkably successful in determining fatigued drivers. To do that, smartphones, and similar devices (e.g., smartwatches and surveillance devices) have been demonstrated to have potential in providing a platform for mental health detection tools to be integrated into. For example, a smartphone-based cortisol detection technique has been introduced to estimate driver fatigue (Russell et al., 2022). This “Smart Fatigue Phone” is a health monitoring technology developed to detect real-time fatigue using immunosensors and a smartphone-linked fluorescence signal reader. The immunosensor functions by measuring the concentration of salivary cortisol levels of an individual. This device has been validated by EEG measurements whereby decreased beta rhythms resulting from mental fatigue correlated with high cortisol levels as measured by the smart fatigue phone (Shin et al., 2019).

However, to improve the accuracy of this device, fatigue related biomarkers (e.g., alpha-amylase, cortisol levels in the blood, and lactate) should be correlated with the salivary cortisol levels. In the sporting environment, it was found that salivary biomarkers were not found to be a good objective indicator of mental fatigue as the changes observed in cortisol and alpha-amylase were not related to mental fatigue (Russell et al., 2022). Another limitation to using salivary cortisol as an assessment tool is that saliva is easily contaminated by blood and food particles in the mouth which can provide inaccurate results of cortisol levels (Kamodyová et al., 2015). This makes testing a little less convenient as individuals would need to rinse their mouths at least 10 min before testing themselves for mental fatigue. On top of that, studies have shown that gender affects the temporal changes in salivary cortisol levels whereby females were observed to have higher cortisol levels than males (Nakajima et al., 2012). According to Montero-López et al. (2018) menstrual cycle phase was found to increase cortisol responses to laboratory induced mental stress. Contrastingly, it was found that in healthy women, the basal cortisol levels were not affected by menstrual cycles (Wilson, Lorenz & Heiman, 2018). These gender effects do not pose as a major limitation to using salivary cortisol as an assessment tool for mental fatigue, but it does make this testing method more complicated, as the assessment tool will need to be calibrated specifically for each gender. Overall, there is no evidence linking this method to a decline in executive functions, specifically inhibition, which studies have shown to be a strong indication of mental fatigue in individuals (Guo et al., 2018).

Saccadic eye movements

In recent years, characteristics of saccadic eye movements have gained popularity in identifying various neurodegenerative-related cognitive deficits such as those found in Alzheimer’s Disease (AD) and Parkinson’s disease (Anderson & MacAskill, 2013; Opwonya et al., 2021). Saccadic eye movements can be described as the rapid movement of the eyes from one point to another (Purves et al., 2001a). Saccades play a fundamental role in the ability to attend to the visual environment. For example, we need saccades to rapidly scan the environment (e.g., before crossing the road) and to be able to identify sudden changes in the environment to avoid danger (Purves et al., 2001b). Reflexive saccades are stimulus driven and involves bottom-up processes, whereas voluntary saccades are goal driven and involves higher-level cortical functions to be executed (Mulligan, Stevenson & Cormack, 2013; Goffart, 2009; Walker et al., 2000). Voluntary saccades involve top-down processing, they are vulnerable to natural aging and less resilient, hence deteriorates as a person ages (i.e., from adulthood to old age) or encounters illnesses or disorders (e.g., migraine and Tourette’s Syndrome) that affect higher cortical brain areas (LeVasseur et al., 2001; Filippopulos et al., 2021; Bafna & Hansen, 2021; Chen et al., 2021; Chen & Machado, 2016).

As saccadic eye movements are related to the neural activity in the frontal lobe, the area of the brain that is mostly affected by mental fatigue (Ishii, Tanaka & Watanabe, 2014; Purves et al., 2001b), it is very likely that the reliability and validity of using saccadic eye movements in detecting mental fatigue would be high (Yamada & Kobayashi, 2018). As mentioned previously, the overactivation that occurs in the frontal lobe when an individual is mentally fatigued results in inability to inhibit responses (Gantois et al., 2020; Guo et al., 2018; Jaydari Fard, Tahmasebi Boroujeni & Lavender, 2019). Hence, inhibition responses can be used as a marker to assess mental fatigue. The “antisaccade task” is designed to assess an individual’s ability to suppress reflexive saccades in order to produce a voluntary saccade (as shown in Fig. 1). It involves high levels of executive control and has been shown to be reliable to assess an individual’s executive functions (e.g., inhibition responses) (Chehrehnegar et al., 2021).

Figure 1 An example timeline of a trial involving an anti-saccade task.

(A) Participants fixate on a central point. (B) A saccade signal appears in the peripheral visual field simultaneous with the removal of the fixation point. (C) A correct antisaccade is generated in the direction opposite to the signal.

There are various physiological measurement tools that can be used to assess saccadic eye movement characteristics such as EOG. EOG is more commonly known to be used in measuring saccades and fixations by analyzing the electrical signals on the skin around the eyes produced by the eye movement (Wang et al., 2019). The electrodes are able to detect changes in the orientation of the eyes based on the amount of muscle movements towards or against the positive or negative electrode poles (Jia & Tyler, 2019). It was proposed that placing the electrodes on the forehead is best for measuring eye movements related to fatigue as this placement detects the electrical signals on the skin that is produced by eye movements with the use of fewer electrodes and less interference with facial activities, compared to when electrodes are placed in the conventional method, that is around the eyes (as shown in Fig. 2) (Zhang et al., 2015; Zheng & Lu, 2017; Heo, Yoon & Park, 2017). Furthermore, electrodes placed at the forehead can be integrated into many wearable devices such as headbands and caps, and thus increasing the usability for the user (Heo, Yoon & Park, 2017).

Figure 2 Conventional electrodes placement for eye movement measurement.

Electrode placement for (A) new proposed EOG and (B) conventional forehead EOG. Red circles represent negative electrodes and green circles represent positive electrodes. The Black circle represents the ground electrode. The vertical (V) and horizontal (H) electrodes are used to detect vertical and horizontal saccades, respectively.

Moreover, there are other tools that have been developed to track saccadic eye movements such as infrared based eye tracking techniques. With the advancement of technology, these infrared sensors might be able to be integrated into webcams, handphone cameras and even surveillance cameras in the near future. Infrared sensors can detect saccades in all directions (e.g., horizontal and vertical saccades), however, horizontal saccades have been shown to be sufficient for mental fatigue detection (Jia & Tyler, 2019). It has been found that individuals experiencing mental fatigue presents a specific pattern of eye movement metrics which include increased saccade latency, blink frequency, and decreased pupil dilation range (Yamada & Kobayashi, 2018; Bafna, Bækgaard & Hansen, 2021; Zargari et al., 2018). These characteristics occur as mental fatigue causes deficits in inhibition of response as well as a decrease in information processing speed. The Wearable Transparent Eye Detection System is an example of a health monitoring technology that uses saccades to detect mental fatigue (Sampei et al., 2016). This device measures the movement of the pupil and blinking from light reflected from the eye. This light is detected by dye-sensitized photovoltaic cells. In the same study, the device has also been validated using the NASA Task Load Index (NASA-TLX), a self-reporting questionnaire used to assess an individual’s estimated mental load that results in mental fatigue. However, this validation study was only conducted on a small sample of five subjects, with no correlations reported between pupil movement, blinking, and mental load for two subjects (De Vries et al., 2004). Therefore, further studies are necessary to identify specific saccadic eye movement characteristics that correlate strongly with other measures of mental fatigue.

Recently, many studies have been carried out to develop a method of tracking saccadic eye movements outside of the clinical environment via smartphone cameras and surveillance devices (Lai et al., 2020; Ansari, Kasprowski & Obetkal, 2021). Appearance-based gaze estimation with the help of convolutional neural networks was found to be successful in accurately decoding saccadic eye movements from facial images and desktop cameras (Ansari, Kasprowski & Obetkal, 2021). This model advances the mental fatigue detection method via saccades as it allows for individuals to be tested for mental fatigue whilst they carry out their day-to-day living. This method, unlike other methods previously mentioned, increases the accuracy and convenience for dynamic saccadic eye movement testing for mental fatigue. Future replication and validation studies would be necessary to validate these saccadic eye movement characteristics with other mental fatigue measures as the data would be useful to integrate this detection method into modern health monitoring technologies.

Conclusions

In summary, the aim of this article was to review and compare existing technological developments used for mental fatigue detection for future technological advancement and validation purposes. Studies showed that the brain areas primarily affected by mental fatigue are the visual cortices and the frontal cortex. Environmental factors such as high-pressure occupations, long working hours, poverty, and job strain causes an individual to develop mental fatigue. As deficits in inhibitory functions have been found to be a good indicator of mental fatigue, it can be used as a marker for early detection of mental fatigue. Given the nature of antisaccade tasks in assessing inhibitory functions, this method has a high potential to be used to objectively detect mental fatigue. Each of the mental fatigue detection methods discussed have their own strengths, weaknesses (see Table 3 for summary), and viability of its integration into health monitoring technologies. However, due to the lack of validity and disadvantages in some of the assessment tools, future research should look into validating mental fatigue detection tools that are able to assess biological markers of mental fatigue such as saccadic eye movements and salivary cortisol levels, in order to provide a concrete assessment of mental fatigue. As for now, a combination of assessment methods should be used to accurately detect mental fatigue in individuals.

Table 3 Pros and cons of each mental fatigue detection method.

Detection method	Pros	Cons	
Self-reporting questionnaires	Multiple brief questionnaires used

Reliability has been tested

Easy to administer

Accessible

	Unable to provide moment-to-moment fluctuations of mental fatigue

Subjective and not immune to response biases

May not be able to accurately reflect humans’ cognitive state at that particular moment

	
Attention network task	Covers three fundamental areas of attention: alertness, orientation, and executive control.

	Long testing period (25–30 min).

Practice effect

Mainly cognitive and for inferring purposes.

Attention network too complex to be assessed using a single task.

	
EEG	Theta wave found to be a reliable primary biomarker for mental fatigue.

Alpha wave found to be a reliable secondary biomarker for mental fatigue.

	Requires laboratory setting to reduce signal noise

Unable to accurately detect mental fatigue while preforming cognitively demanding tasks

	
Heart rate variability	Easily integrated into smart devices.

Involuntary biomarker

	Too many confounding variables.

Low reliability

	
Cortisol levels	Can be used alongside health monitoring technologies such as Smart Fatigue Phone

	Lack of evidence to support link between salivary cortisol and mental fatigue

Saliva samples easily contaminated

	
Saccadic eye movements	Reflects brain activities associated with mental fatigue

The saccadic task is simple and quick

Measures voluntary and involuntary responses

Can be detected via EOG or infrared

Can be integrated into health monitoring technology such as smartphones, webcams and recording devices.

	Limited validity of saccadic eye movement characteristics linked to mental fatigue.

	

Additional Information and Declarations

Competing Interests

Author Contributions

Data Availability

The authors declare that they have no competing interests.

Kaveena Kunasegaran conceived and designed the experiments, performed the experiments, analyzed the data, prepared figures and/or tables, authored or reviewed drafts of the article, and approved the final draft.

Ahamed Miflah Hussain Ismail analyzed the data, authored or reviewed drafts of the article, and approved the final draft.

Shamala Ramasamy analyzed the data, authored or reviewed drafts of the article, and approved the final draft.

Justin Vijay Gnanou analyzed the data, authored or reviewed drafts of the article, and approved the final draft.

Brinnell Annette Caszo analyzed the data, authored or reviewed drafts of the article, and approved the final draft.

Po Ling Chen conceived and designed the experiments, analyzed the data, authored or reviewed drafts of the article, and approved the final draft.

The following information was supplied regarding data availability:

This work does not involve raw data or code.

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
