# Peer review of "Understanding mental fatigue and its detection: a comparative analysis of assessments and tools"

_PeerJ, doi:10.7717/peerj.15744_

## Round 0.1 · original submission · Major Revisions

Please address the issues highlighted by the Reviewers and make the corresponding changes to the manuscript. This will improve the manuscript substantially.

Reviewer 1 ·

Basic reporting

The authors present an introduction and review of various behavioral and psychophysiological measures of mental fatigue. An introduction is first given what mental fatigue is, how it is different from stress, what brain areas are affected, how it is caused, and how it affects cognitive and physical performance. Subsequently, a number of measures is discussed how mental fatigue can best be detected, including questionnaires, an attention test, EEG, heart rate variability, cortisol levels, and saccadic eye movements.

The authors provide an interesting overview of a number of techniques that are currently used or provide much promise to detect mental fatigue, giving a basic introduction to the field. Unfortunately, I do feel though that the current manuscript has quite some room for improvement, as detailed below.

Main points:
1. I miss a clear rationale for the review. What does this add to previous reviews on the topic, such as Aaronson et al 1999; Martins & Carvalho, 2015, Hu & Lodewijks, 2020, Monteiro et al, 2019; Tran et al, 2020; Bafna & Hansen, 2021? Please be more explicit explaining the structure and goals of the review, both in the abstract and the introduction.
2. I was regularly surprised/confused by the structure and order of topics, and the discussion of each topic was sometimes quite shallow or incomplete. For instance, I would expect a discussion of ‘environmental causes of mental fatigue’ before ‘brain mechanisms’. Also, I would expect ‘self-reporting questionnaires’ and subsequent measures as subsections of ‘mental fatigue detection’.

Experimental design

3. The absence of a discussion of any fMRI or fNIRS studies on the topic of mental fatigue.
4. Only the ANT was discussed, whereas a discussion of performance on attention tasks such as the Psychomotor Vigilance Task (Dinges & Kribbs, 1991) or the Eriksen Flanker Task (Eriksen & Eriksen, 1974) that typically show effects of mental fatigue are missing. On line 513, the authors write that “Mental fatigue affects visual attention, physical performance, and cognitive performance.” Not only the order in which the topics are mentioned surprised me, but I also had trouble linking the concepts of visual attention and cognitive performance to specific sections in the manuscript. Or are both measured by the ANT test? Should the ANT be seen as an attention measure or a measure of executive function (as indicated in table 3)? Please clarify.

Validity of the findings

No comment

Additional comments

Minor points:
5. The writing should be checked by a fluent English speaker. Some examples of places that need attention:
• Please rephrase the manuscript’s title
• Line 36: “These are the rationales motivate this review,” please rephrase. Also it remains unclear what ‘These are the rationales’ refer to.
• Line 159: magnetoencephalogram -> magnetoencephalography (MEG)
• Line 186. Long working hours can cause general fatigue, not individuals working long hours.
• Line 306-307: a potential -> potential
• Line 316-317: please rephrase
• Line 379: please remove ‘more’
• Line 391: ‘…a more reliable indicator’ Compared to what? Please remove ‘more’
• Line 424: remove ‘for’?
• Line 468: lesser -> fewer
• Table 3: assess -> assessed

6. Lines 129-143: It is unclear whether and how the concept of passive fatigue differs from the concept of sleepiness (see Hu & Lodewijks, 2020). Also, please clarify that you will focus on the active state of fatigue in this review.
7. Lines 308-312: It is rather odd to criticize a behavioral test for not being a psychophysiological test.
8. Line 352: ‘the most sensitive measure’ Compared to what?
9. Lines 466-472: It remains unclear to me why/how the new way of placing the electrodes detects 6 rather than 3 types of eye movements, and why this is important given that horizontal movements seem sufficient for the detection of mental fatigue, as mentioned on lines 479-480.
10. Line 434: please add an appropriate reference here.
11. Table 1: How can the FAS be unbiased concerning gender and have 4 out of 10 items that show a gender bias at the same time?

Reviewer 2 ·

Basic reporting

I read the reviewed paper with interest. The paper provides an overview of current methods of assessing mental fatigue while outlining their strengths and weaknesses. Despite the use of many physiological parameters, we still do not have a simple and reliable method of measuring mental fatigue, especially one that can be used directly at workplaces. Hence the continued need to use psychological scales and questionnaires.
I have no comments on the reviewed paper except for a few editorial comments – below.
L. 114 – should be ‘’fight-or-flight”.
L. 514-515 – this sentence should be rephrasing.
L. 635, 653, 708, 775, 798, 810, 929, 951, 966, 1045 – lack information regarding volume number, page range or publisher.
L. 891 – should be …Heart Circ Physiol
Table 1 and 2 - The numbering of the papers cited in the tables should be consistent with the numbering from the References.

Experimental design

No comment.

Validity of the findings

No comment.

---

## Round 0.2 · accepted · Accept

Authors have adapted the manuscript to address all of the reviewers' comments. The manuscript is ready for publication.